# Assessment of Stickiness with Pressure Distribution Sensor Using Offset Magnetic Force

**DOI:** 10.3390/mi10100652

**Published:** 2019-09-27

**Authors:** Takayuki Kameoka, Akifumi Takahashi, Vibol Yem, Hiroyuki Kajimoto, Kohei Matsumori, Naoki Saito, Naomi Arakawa

**Affiliations:** 1Department of informatics, The University of Electro-Communications, Tokyo 182-8585, Japan; a.takahashi@kaji-lab.jp (A.T.); yem@kaji-lab.jp (V.Y.); kajimoto@kaji-lab.jp (H.K.); 2Shiseido Global Innovation Center, Kanagawa 220-0011, Japan; kohei.matsumori@shiseido.com (K.M.); naoki.saito@shiseido.com (N.S.); naomi.arakawa1@shiseido.com (N.A.)

**Keywords:** haptics, measurement techniques, stickiness, sticky feeling

## Abstract

The quantification of stickiness experienced upon touching a sticky or adhesive substance has attracted intense research attention, particularly for application to haptics, virtual reality, and human–computer interactions. Here, we develop and evaluate a device that quantifies the feeling of stickiness experienced upon touching an adhesive substance. Keeping in mind that a typical pressure distribution sensor can only measure a pressing force, but not a tensile force, in our setup, we apply an offset pressure to a pressure distribution sensor and measure the tensile force generated by an adhesive substance as the difference from the offset pressure. We propose a method of using a magnetic force to generate the offset pressure and develop a measuring device using a magnet that attracts magnetic pin arrays and pin magnets; the feasibility of the method is verified with a first prototype. We develop a second prototype that overcomes the noise problems of the first, arising from the misalignment of the pins owing to the bending of the magnetic force lines at the sensor edges. We also obtain measurement results for actual samples and standard viscosity liquids. Our findings indicate the feasibility of our setup as a suitable device for measuring stickiness.

## 1. Introduction

Haptic perception has begun garnering increasing attention over the past few years, and in this regard, several studies have examined the representation of the human skin sensation, particularly in the fields of virtual reality and human–computer interactions. In reproducing a realistic feeling, it has been found effective to measure changes in the skin condition, such as the skin deformation distribution and contact area, in real-world situations and to reproduce this information. For example, Levesque et al. [1] measured the horizontal displacement of the skin of a finger tracing a glass surface in detail to capture information on how the skin deforms when in contact with an irregular shape. Bicchi et al. [2] captured changes in the skin contact area for a finger touching a flexible object. Such measurements are closely related to the technique used in tactile presentation [3]. The measurement of the horizontal displacement of skin is related to the development of devices that present horizontal displacement [4,5] and the measurement of the skin contact area has led to the development of devices that can represent a feeling of flexing/flexibility via changing the contact area [6,7].

Against this backdrop, here, we focus on the distribution of skin deformation corresponding to the feeling of “stickiness”. In this study, stickiness is defined as the feeling experienced when touching an adhesive material such as glue, Nattō (which is a traditional Japanese food made from soybeans), or honey. The feeling of sticky sensation on the surface is also expressed as a frictional resistance [8]. Chen et al. [9]. investigated the correspondence between the measurement of physical properties of texture surface and subjective evaluation of touch sensation, and also mentioned the stickiness. However, we address stickiness that is related to the motion of the finger along surface normal, that is, the sensation experienced after releasing a finger that has been pressed against a sticky material. Here, we note that stickiness affects our impression of daily consumables such as lotion, creams, and so on. Moreover, stickiness is known as one of the factors responsible for the feeling of “wetness” in fabric perception [10,11]. Stickiness is often used as a general aversion sensation [12] and is a quality that is attributed to a wide range of materials and products. 

Though Liu et al. measured adhesive force by the MEMS device [13], the development of the system corresponding to the adhesive force measurement on the skin is necessary in order to evaluate the sense of human. It has been speculated that both proprioceptive and cutaneous sensations contribute to stickiness; here, however, we mainly focus on cutaneous sensation. In this context, Yamaoka et al. [14] derived the relationship between the contact area of an adhesive surface and the temporal change in the pressing force, and found that there is large hysteresis in the contact area. The authors further created a stickiness display based on this finding. However, because their observations were limited to the change in the contact area, the detailed force distribution during the period of stickiness was not clearly elucidated. Such detailed physical properties are often required to be measured in the field of food [15,16]. For example, Dan et al. [17]. measured the bite force applied to raw and cooked apples using a pressure distribution sensor sheet.

In previous reports, we have described the basic principle of a system that measures the force distribution between adhesive substances and finger skin [18,19]. We used a pressure distribution sensor sheet to measure the adhesive force in the form of pressure distribution. Here, it must be noted that common pressure distribution sensors can measure only positive (compressive) pressure, but not negative pressure. Therefore, we devised a method of inserting a pin matrix between the skin and the pressure distribution sensor to apply a “preload” using the weight of the pins. With this configuration, the adhesive force can be observed as a decrease in the offset force when the finger is raised. In our previous studies, the weight of the pin was set to 0.8 g, and in the case of a highly sticky specimen, the pin could float because of the stronger adhesive force. Therefore, it was necessary to apply a stronger preload to the sensor to perform more stable measurements. The other issue in our previous studies involved the sensor sheet. Pressure-sensitive rubber sensors are prone to undesired current pathways and large hysteresis, both of which make it difficult to realize accurate sensing using prototype systems. 

In this paper, we present and evaluate a stickiness measurement device with a large measurement range. When compared with our previous approaches, here, we used a load cell substrate with independent sensing points and applied a more powerful preload using a magnetic force.

## 2. Measuring System

### 2.1. Pressure Distribution Sensor Using Load Cell

To measure a distributed adhesive force, we developed a pressure distribution sensor using load cells. Figure 1 shows the load cell, schematic, and photo of sensor substrate. In our study, we used HSFPAR003A load cells (Alps, Inc. Tokyo, Japan) to measure the pressure; this load cell allows the measurement of forces up to 3.5 N. Sensing points are located at 2.54 mm intervals, which correspond to the two-point discrimination threshold of the human fingertip [20]; this interval is thus sufficient for measurement. One-unit board has 16 load cells. One load cell is selected and amplified by analog multiplexer (ADG726BCPZ, Analog Devices, Norwood, MA, USA) and differential amplifier (AD623ARMZ, Analog Devices, Norwood, MA, USA).

### 2.2. Offset Pressure Generated by Magnetic Force

Figure 2 shows the overall measurement system. The device consists of an acrylic pin insertion plate, an 8 × 12 magnetic pin array, three base magnets (50 mm × 50 mm × 10 mm), six load cell substrates (each mounted with 4 × 4 load cells), and an acrylic fixed pedestal. 

In this study, to apply an offset preload to the load cell, a magnet was installed under the load cell substrate and a pin matrix made of pin magnets was inserted between the point of skin contact and the load cell substrate (Figure 2). Each pin of the magnet pin array was aligned to correspond to a single load cell. With this configuration, because the magnet under the substrate and the pin magnet matrix attract each other, an offset pressure (i.e., preload) can be applied to the load cell. Three magnets were stacked to strengthen the magnetic field (which was about 430 mT at the center). The magnetic field of adjacent pins might interfere with each other. However, because the polarities of the pin magnets are all set in the same direction, and the pin magnets are all at the same height and approximately horizontal, the force generated by the interference is the repulsive force in the tangential direction, which, in principle, has no effect on the normal force measurement. The load cells and magnet pin array were both positioned at 2.54 mm intervals. The pin magnet was 2 mm in diameter and 10 mm in height, and the magnetic force was 275 mT. The acrylic plate and the pedestal were made with a laser cutter. The acrylic plate was chosen because it is easy to cut using the laser cutter. The cut surface becomes slightly conical, and the contact area with the pin becomes smaller, reducing the friction.

We used six load cell substrates mounted on the base board, constituting 96 measurement points. The voltage signal from a single load cell is selected by the analog multiplexer, amplified by the differential amplifier, and then measured by an AD converter (MCP3208, Microchip Technology Inc., Chandler, AZ, USA). All operation is conducted by a micro-controller (mbed LPC1768, NXP Semiconductors N.V., Eindhoven, Netherland), and the data are sent to PC via USB serial port. Measurements of all points were conducted 60 times per second, and simple moving averages were calculated for noise removal (window size was 16). 

We calibrated the load cell by adding a known weight on load cell. Figure 3 shows the result of single unit, showing high linearity (R^2^ > 0.99) and the load cell value was 10.5 per 1 g. That is, the measured weight is 0.095 gf (=0.94 mN) per value 1 of the load cell. It also shows a large offset value (i.e., the output value was about 700 out of 4095 of the resolution of MCP 3208 without weight) owing to magnetic force, which corresponds to around 686 mN preload, which is sufficient for daily tangible materials such as foods and cosmetics. Figure 4 shows the magnetic field distribution of the magnet under the substrate, measured at the surface of the base magnet at 6.25 mm intervals using TM-801 tesla meter (KANETEC, Inc., Nagano, Japan). Although the magnetic force of the magnet under the substrate varies depending on the measurement position, we note that a sufficient preload ranging from 686 to 1323 mN can be exerted at any cells. In other words, this system cannot measure adhesive substances with adhesive force of more than 686 mN at a single pin. While each pin receives the different magnetic force, this value is treated as an offset and the pressure change amount can be measured as a relative value regardless of the initial offset value.

### 2.3. Preliminary Experiment

In our stickiness measurement experiments, the adhesive material of interest was applied to a participant’s fingertip, who then pressed the fingertip onto the surface of the pin array. When the finger pressure reached 5 N (summed force over all pins), the participant was asked to release the fingertip along the vertical direction. The lifting process was completed in around 1 s. In the study, we used Nattō stirred for 100 times by chopsticks as an adhesive material and baby powder (Johnson & Johnson, Inc. Tokyo, Japan) as a non-adhesive material. Natto is a fermented food in Japan, and when mixed, it becomes sticky.

### 2.4. Results and Discussion

Figure 5 and Figure 6 show the measurement results for Nattō and baby powder, respectively. Although we acquired 2D distributed data, representation by 3D graph was not easy to grasp and we chose to show the 2D view, longitudinal section of the center. As the measurement points are 8 by 12, the horizontal axis of the graphs becomes 1 to 12. The vertical axis represents force (mN), with a positive value meaning tensile force (i.e., negative pressure). In the case of Natto, pressing begins from 0.00 s and finger detachment begins after 2.67 s, totally detached at 4.01 s. In the case of baby powder, pressing begins from 0.00 s, finger detachment begins at 1.67 s, and totally detached at 2.67 s. We asked the participant to release his finger within 1 s, and we visually confirmed that the finger had separated around this time.

Upon comparing the two figures, we note that peak tensile force is stronger in the case of Nattō. From 2.67 s, it can be observed that the tensile force is generated from the periphery of the surface being pressed and gradually concentrated at the center. In both cases, after the finger was totally detached, we still observe remaining tensile force, which is considered as noise. 

One possible explanation of this offset noise is that the pin magnets were aligned along the magnetic lines of the base magnet and the lines were not strictly vertical, which gave rise to the interference between the pin and the pin insertion plate, which may have generated friction and thus noise (Figure 7). This noise can be regarded as a hysteresis of the measuring device.

## 3. Improvement of Measuring System

To solve the problem of the previous prototype—in which the pin magnets were aligned along the base magnet’s magnetic lines of force and did not stand vertically, thus potentially causing friction and noise—we next devised a one-row pin-array measuring device. This device is based on the principle that when an infinitely long rectangular magnet is used, the magnetic force line at its center is vertical (Figure 8). Consequently, in the improved device, the pin magnets are installed vertically on the centerline of the rectangular magnet, which reduces the interference with the pin insertion plate (Figure 9). Although only one row can be measured using this configuration when the object in contact with the adhesive substance is semicircular, the distribution of the adhesive force is concentric and measurement at one row is thus sufficient. As we use a human fingertip or artificial human finger here, which can be considered semicircular, we consider a line measurement to be sufficient.

### 3.1. Measuring System Using Single-Axis Robot

To apply the offset preload to a load cell, we installed a rectangular magnet such that the center line of the rectangular magnet overlapped with the sensor portion of the single-row load-cell substrate. As a result, the pin magnets were positioned along the vertical lines of the magnetic force. In addition, two rows of pin magnets were added so as to sandwich the measuring pins, although actual measurement was performed only for the pin magnets of the center row. These additional rows of pins push the central pins from both sides with magnetic repulsive force, making the central pins perpendicular to the base surface and thus minimizing friction between the pin insertion plate and the pins. The actual device is shown in Figure 10. The device consists of an acrylic pin insertion plate, a 3 × 16 magnet pin array, two 100 mm × 20 mm × 7 mm magnets, and a 1 × 16 load cell substrate. Two base magnets were stacked to strengthen the magnetic field, which was about 410 mT at the center. The load cell and its surrounding circuit components are the same as in the previous setup, while the load cell substrate was redesigned to achieve 1 × 16 load cell configuration. Each pin of the magnet pin array was aligned to correspond to one load cell, and a preload was applied to the load cells.

The load cells were arranged at intervals of 2.54 mm, and the magnet pin array was arranged in the same manner. Each pin magnet was 2 mm in diameter and 10 mm in height, and the magnetic force was 275 mT, corresponding to a preload of about 980 mN.

For contact with the adhesive substance, we used a hemispherical artificial human skin gel (with a diameter of 5 cm, manufactured by BEAULAX Corporation, Saitama, Japan) with elasticity equivalent to that of human skin as the contactor along with a single-axis robot (T4L manufactured by YAMAHA, Shizuoka, Japan) to depress the artificial “finger”. 

### 3.2. Experiment 1

Adhesive substances were uniformly applied to the upper part of the pin array beforehand, and the contactor was pressed against the upper surface of the pin array placed on the load cell. We placed 1 mm thick adhesive materials on each pin, with the procedure depicted in Figure 11. The temporal change in the pressure distribution was measured when pulling apart the contactor from the surface. The single-axis robot was used for pressing and separating the “finger”, and the pushing distance of the contactor was set to 2 mm vertically downward from the state in contact with the pin array, while the pulling-off distance was set to 2 mm vertically upward from the state in contact with the pin array. The moving speed of the contactor was 1 mm/s. Honey, toothpaste, shaving gel, and shampoo were prepared as adhesive substances. These adhesive substances are familiar to us, and we can find difference by touching them. If this system can measure the difference in adhesive force of these substances, there is a possibility that the system can grasp the difference in the adhesive feeling felt by human skin. The sticky substance attached to the contactor and pin arrays was removed each time before the next trial.

### 3.3. Results and Discussion

The measurement results in the case of honey are shown in Figure 12. The horizontal axis represents the pin location, whereas the vertical axis represents the force. A positive value indicates a tensile force (i.e., negative pressure). From the figure, we note that pushing starts at 0 s, whereas pulling begins at around 3.00 s. A tensile (i.e., adhesive) force is observed at the edge of the contact surface. It can also be confirmed that the adhesive force transfers to the center as the contact surface area changes with the motion of the contactor. Overall, the noise was reduced from the previous prototype, and we now can clearly observe tensile force behavior.

On the other hand, we still have some issues. When comparing the pressure distribution after measurement (5.00 s) and before measurement (0 s), we observed a residual force in the positive direction after the measurement. The reason for this hysteresis phenomenon is unclear (we have confirmed that the load cell itself does not have observable hysteresis), but we presume that friction between the contact pin and the plate still existed. Further, a tensile force can be observed to the leftmost section of the graph in the interval from 1.33 s in Figure 12, which should not have contacted the contactor and should be regarded as noise. As the rightmost and leftmost sensing points are not surrounded by other pin magnets, they experience a magnetic force from the neighboring pin magnets to generate a repulsive force in the left and right directions and interfere with the pin insertion plate.

Figure 13, Figure 14 and Figure 15 show the measurement results for toothpaste, shaving gel, and shampoo, respectively. As in the case of honey, we were able to measure the change in adhesion. When the results were compared for each adhesive sample, it was found that in the case of toothpaste, the adhesive force was the strongest and that the viscosity was high and the duration of the adhesive force was long, together with the fact that the adhesive force remained up to 5.0 s. The shaving gel and shampoo had weak adhesion.

### 3.4. Experiment 2

Subsequently, in order to verify whether the difference in adhesive force could be measured by this system, we carried out measurements using a standard viscosity liquid. In this experiment, kinematic viscosities (Centi-Stokes Visco Liquid, ASONE, Inc., Tokyo, Japan) of 500, 1000, 3000, 5000, and 10,000 (cSt) were measured. The moving speed and force of the contact object in the measurement were set to values identical to the corresponding ones in experiment 1. The peak adhesive force during one measurement was acquired 10 times consecutively for each sample, and the average value was used as the measurement data of each sample.

### 3.5. Results and Discussion

Figure 16 shows the peak value of the adhesive force for each sample as acquired from the measured data. As can be observed from the figure, there is an obvious correlation between the peak values of the adhesive force and kinematic viscosity, and the difference in the adhesive force can be suitably measured by means of our measuring device. The peak value of the adhesive force becomes constant when the kinematic viscosity exceeds 5000 cSt. 

The purpose of fabricating the second prototype was to solve the issues of the first prototype, specifically the large observed noise possibly owing to friction between the pins and the plate caused by inclined magnetic lines. Our results suggest that this noise is reduced with the new setup, and the tensile force distribution can be clearly observed. This in turn implies that our speculation of the cause of the noise was correct.

## 4. Conclusions

In our study, we developed a measuring device for quantifying stickiness. A pressure distribution sensor was used to observe the temporal change in the pressure distribution upon applying finger pressure to a given adhesive material. Here, we note that a typical pressure distribution sensor can measure a pressing force, but not a tensile force, and thus we proposed and implemented a method of measuring the tensile force by applying an offset pressure in advance to the sensor and measuring the resulting difference upon finger pressure application.

Subsequently, we developed a pressure distribution sensor board with built-in load cells and independent sensing points for highly accurate measurements and implemented a preload application method using a magnetic force. We next compared measurements acquired using a fingertip coated with Nattō as an adhesive and a fingertip coated with baby powder. The results confirmed that the tensile force generated by the adhesive substance was initially at the edge of the contact part, but moved with the change in contact pressure, and eventually converged to the center.

However, owing to the inclination of the magnetic lines of force, there was interference between the pin magnet and the pin insertion plate. Therefore, we proposed a linear arrangement to simplify the system and provide a stable vertical preload. We subsequently measured several kinds of adhesive substances using a single-axis robot that pressed an artificial finger onto the device coated with the sticky substance of interest. The results indicated that the new setup was able to measure the adhesive force distribution more accurately. We believe that our device can be used to suitably quantify the stickiness of adhesive substances. 

The current obvious limitation is that we did not fully eliminate noise. Our method also has some innate drawbacks, such as the fact that it cannot measure adhesive force exceeding the offset pressure, and excessively low viscosity substances such as water cannot be treated because they fall from the magnet pins. Still, we believe that the measurement of adhesive force distribution should contribute to the study of stickiness, and comparing our measured data with a human’s subjective tactile feeling is our next step.

## Figures and Tables

**Figure 1 micromachines-10-00652-f001:**
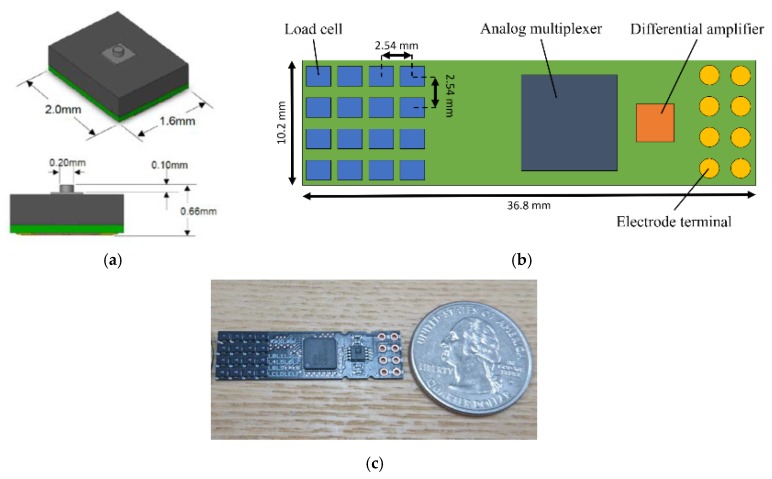
(**a**) Load cell used for pressure sensing. (**b**,**c**) Schematic and photo of sensor substrate comprising 4-by-4 load cells.

**Figure 2 micromachines-10-00652-f002:**
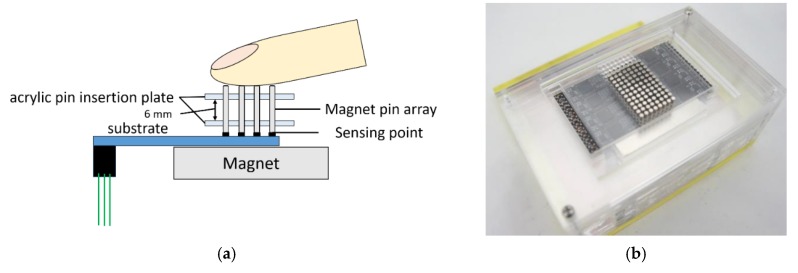
Measurement system based on magnet (**a**) structure, (**b**) oblique view, (**c**) base magnet, and (**d**) circuit board of sensing device. The attractive force between the base magnet and the pin magnets works as a preload to the load cell at the sensing point, enabling measurement of tensile force distribution of adhesive material. Six load cell substrates are mounted on the base board, constituting 96 measurement points.

**Figure 3 micromachines-10-00652-f003:**
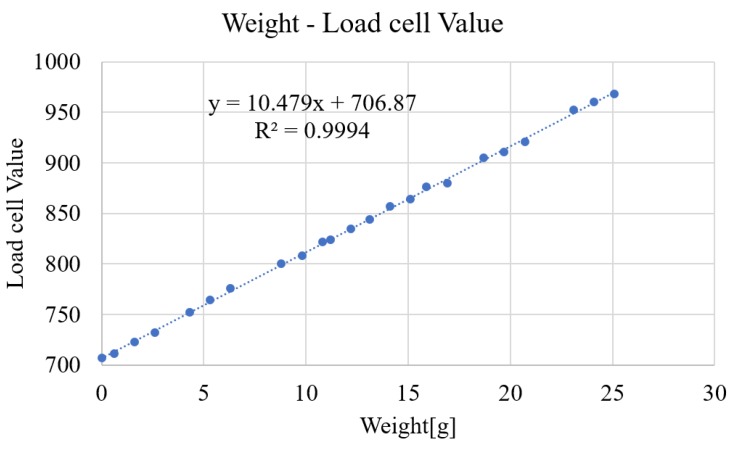
Load cell calibration. We measured load cell value by applying offset pressure by the magnetic force and arbitrary weight.

**Figure 4 micromachines-10-00652-f004:**
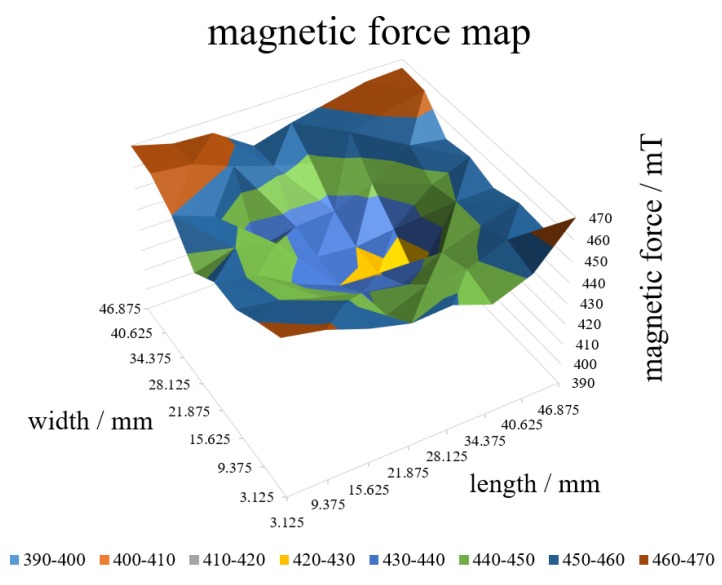
Magnetic force map. The magnetic force was measured at intervals of 6.25 mm along the length and width directions immediately at the surface of the magnet.

**Figure 5 micromachines-10-00652-f005:**
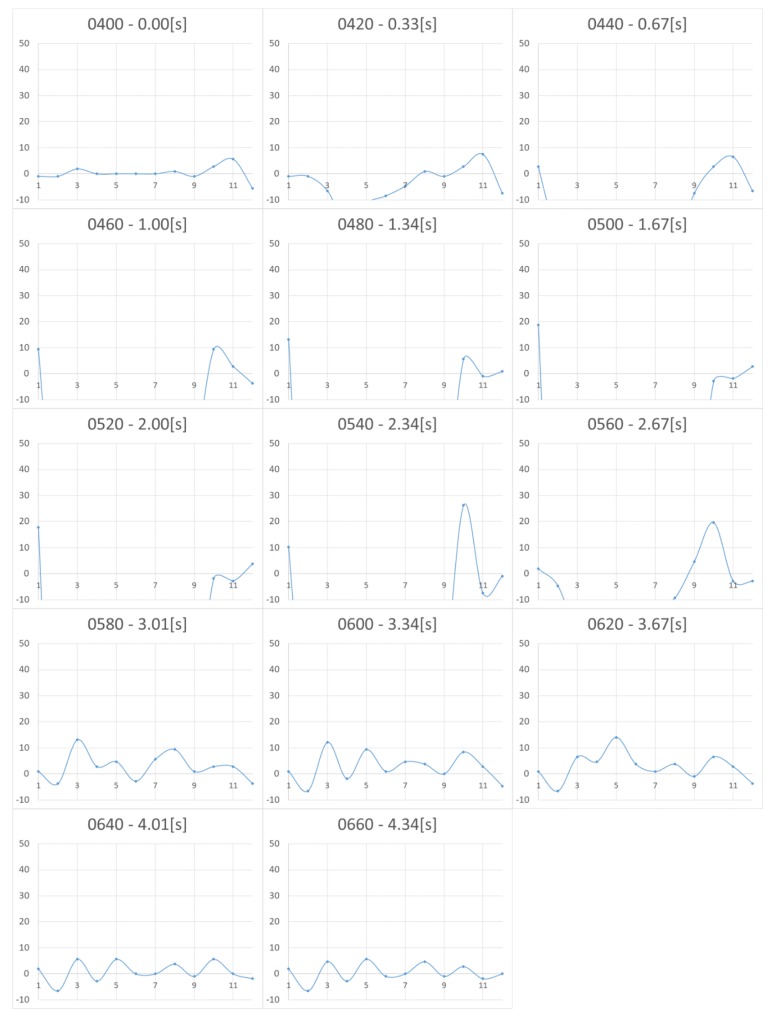
Change in pressure distribution for Nattō (2D view, longitudinal section of the center). The vertical axis represents the force (in milli Newton). The horizontal axis represents the location of the sensing point. The four-digit numbers in each graph show the frame numbers at the time of measurement. Measurements were taken in 0.0165 s/frame (60 fps).

**Figure 6 micromachines-10-00652-f006:**
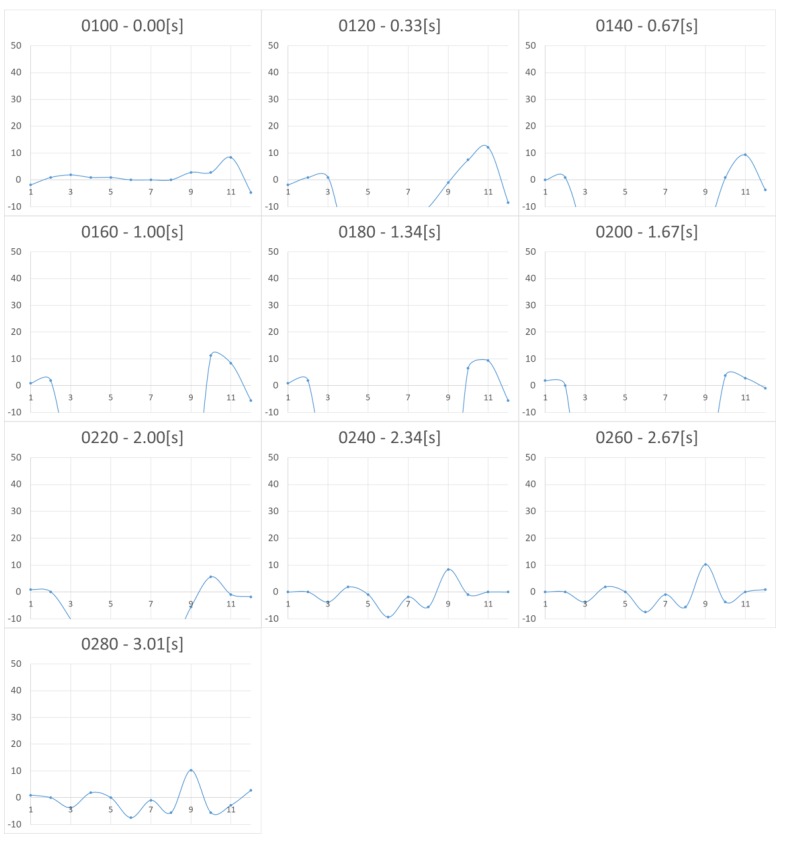
Change in pressure distribution for baby powder (2D view, longitudinal section of the center). The vertical axis represents the force (in milli Newton). The horizontal axis shows the location of the sensing point. The four-digit numbers in each graph show the frame numbers at the time of measurement. Measurements were taken in 0.0165 s/frame (60 fps).

**Figure 7 micromachines-10-00652-f007:**
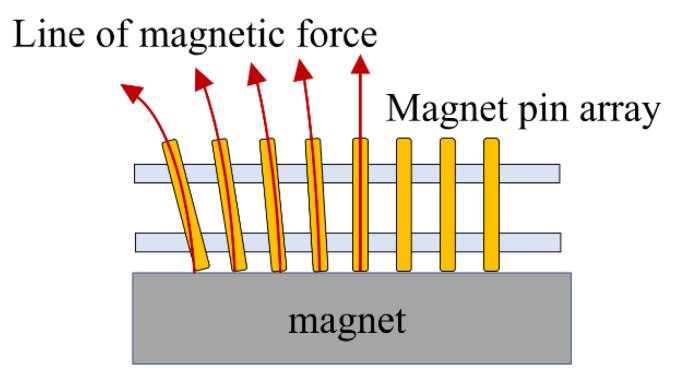
Magnetic lines of force in the setup. The magnetic field lines of the base magnet have a tangential component, generating frictional force between the pin magnet and the pin insertion acrylic plate.

**Figure 8 micromachines-10-00652-f008:**
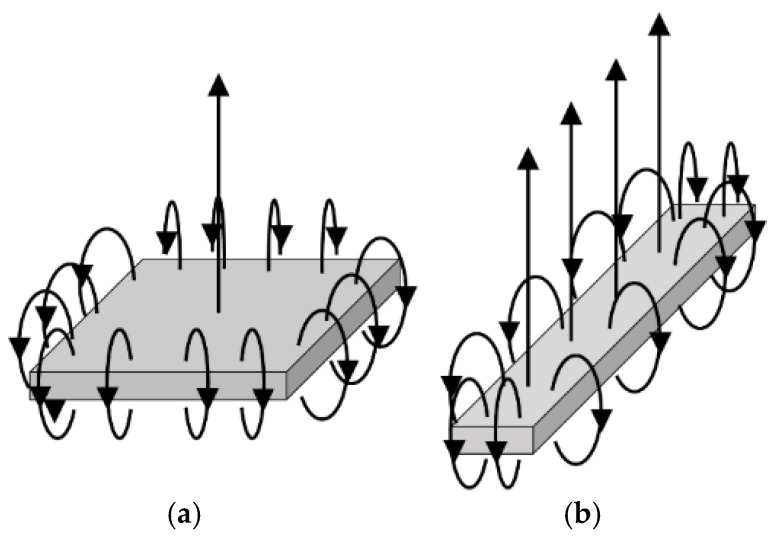
Magnetic field lines of square (**a**) and rectangular magnets (**b**). When an infinitely long rectangular magnet is used, the magnetic force line at its center is vertical.

**Figure 9 micromachines-10-00652-f009:**
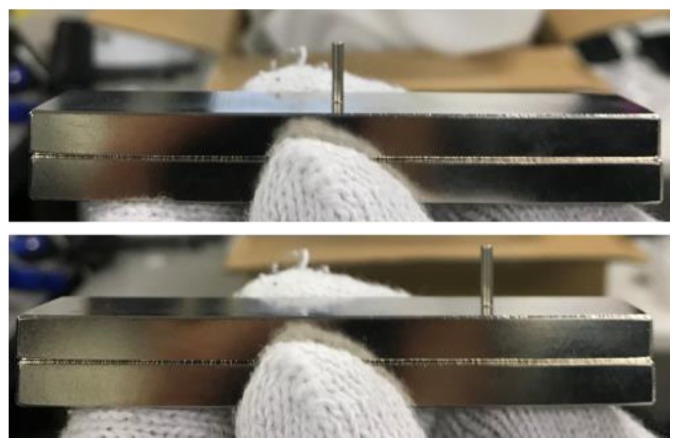
Pin magnet positioned at center and peripheral regions of rectangular magnet. The photographs indicate that the pin can remain vertical because of vertical magnetic lines.

**Figure 10 micromachines-10-00652-f010:**
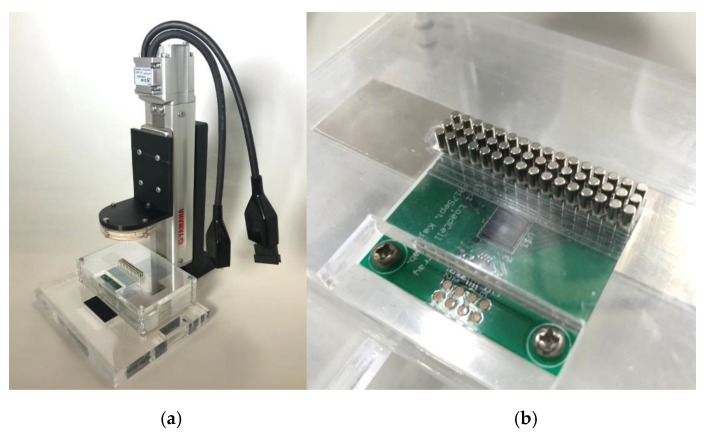
Single-column pin-array measuring device (**a**) overall view, (**b**) magnified view of the measurement part.

**Figure 11 micromachines-10-00652-f011:**
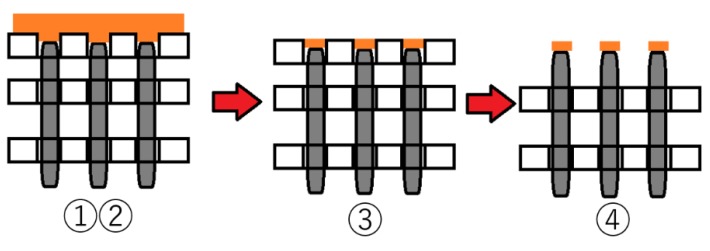
Procedure of pasting adhesive material on each pin. ① Place the insertion plate over the pin magnet, ② coat with an adhesive sample, ③ remove excess adhesive samples, and ④ remove the insertion plate.

**Figure 12 micromachines-10-00652-f012:**
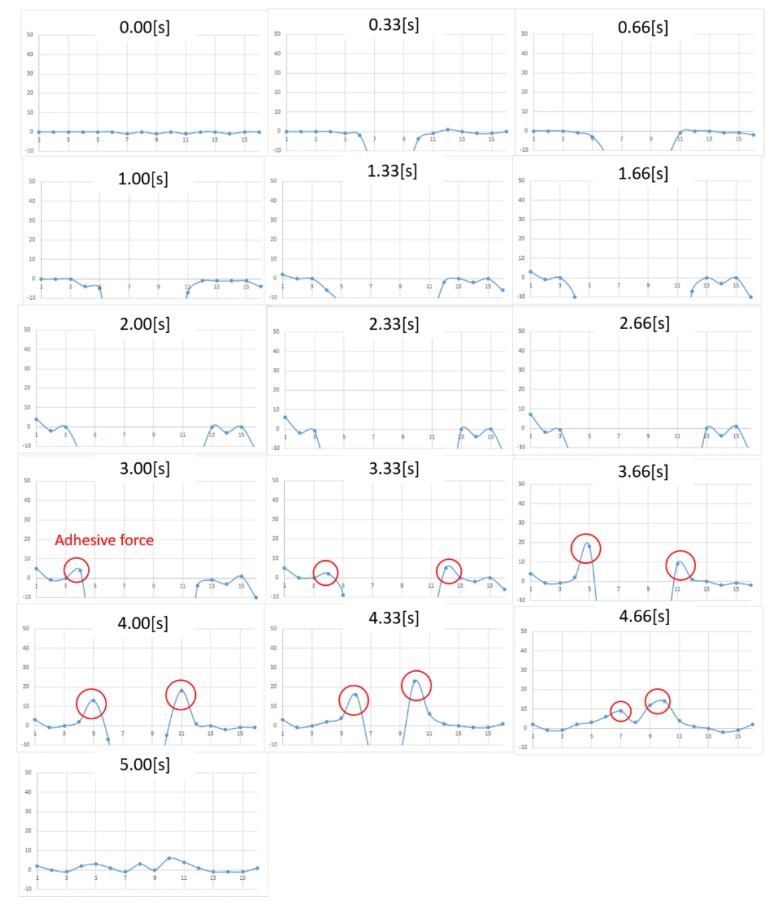
Change in pressure distribution for honey. The vertical axis represents the force (in milli newton). The horizontal plane represents the location of the sensing point.

**Figure 13 micromachines-10-00652-f013:**
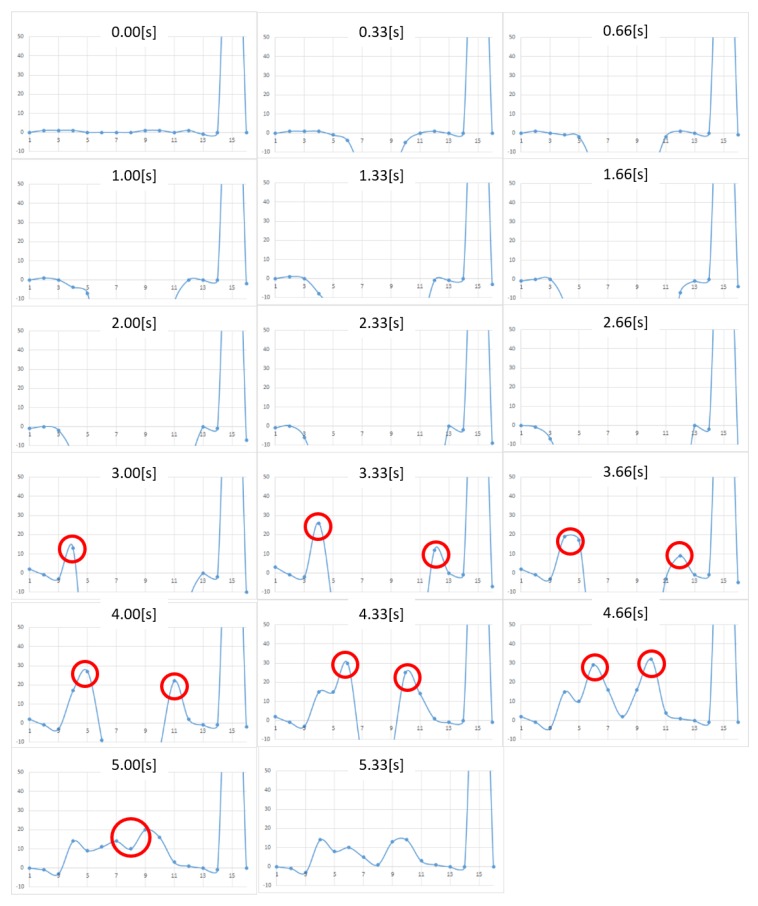
Change in pressure distribution for toothpaste. The vertical axis represents the force (in milli newton). The horizontal plane represents the location of the sensing point.

**Figure 14 micromachines-10-00652-f014:**
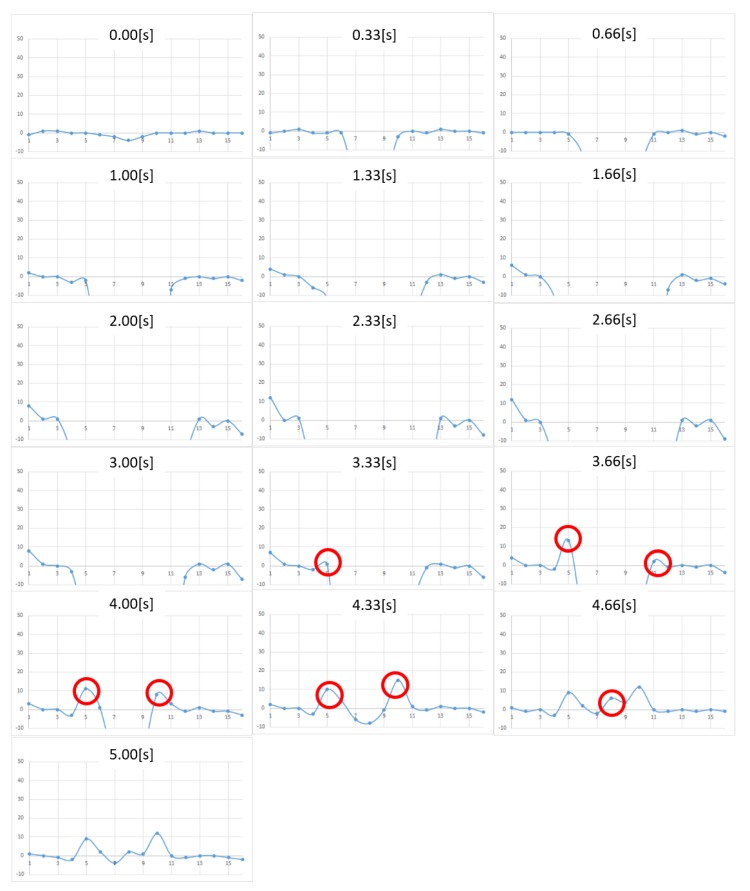
Change in pressure distribution for shaving gel. The vertical axis represents the force (in milli newton). The horizontal plane represents the location of the sensing point.

**Figure 15 micromachines-10-00652-f015:**
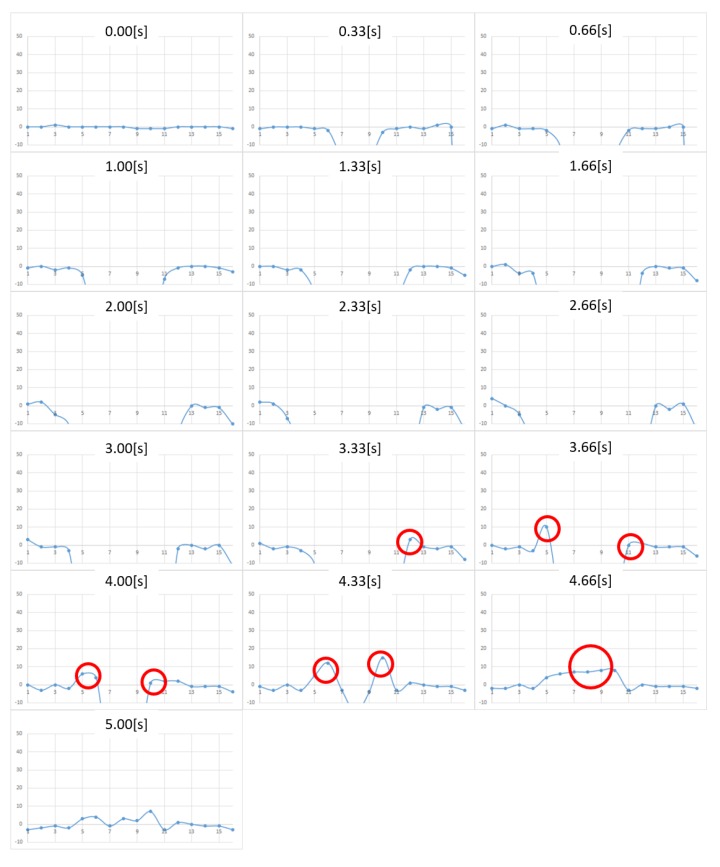
Change in pressure distribution for shampoo. The vertical axis represents the force (in milli newton). The horizontal plane represents the location of the sensing point.

**Figure 16 micromachines-10-00652-f016:**
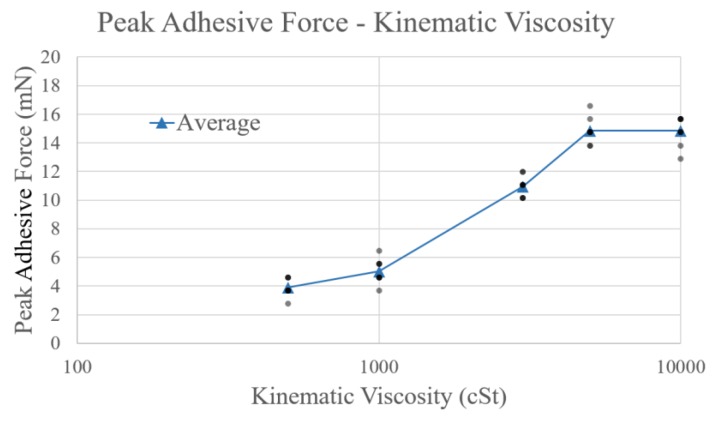
Plot of peak values of adhesive force (mN) and kinematic viscosity (cSt). The blue line indicates the average peak value, whereas the gray points indicate the row data (for standard viscosity liquids of 500, 1000, 3000, 5000, and 10,000 cSt).

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
