# Peer review of "Assessment of Stickiness with Pressure Distribution Sensor Using Offset Magnetic Force"

_micromachines, 2019, doi:10.3390/mi10100652_

Round 1

Reviewer 1 Report

The authors addressed most of my concerns.

* Line 151. Unfortunately, I still cannot follow what you mean by Natto been stirred 100 "times". What do you mean by "times"? 100 rotations with a tool/mixer/spoon?

* There is a reference error on line 155.

Author Response

Response to the comments of Reviewer 1

Comments and Suggestions for Authors

The authors addressed most of my concerns.

* Line 151. Unfortunately, I still cannot follow what you mean by Natto been stirred 100 "times". What do you mean by "times"? 100 rotations with a tool/mixer/spoon?

Thank you for your comment. We used chopsticks to stir the Natto 100 times. We added description about this in lines 154.

* There is a reference error on line 155.

Thank you for your insight. We fixed this mistake.

Reviewer 2 Report

The authors designed a stickiness measuring apparatus consist of a load-cell array and pre-loading mechanism using magnets. The key idea is applying positive offset force to measure the negative pull from adhesion. The clear advantage of having a 2D array of sensors is the ability to detect force distribution.

I like the design, working principle, and the fabrication of the sensor. However, the validation experiments have a room for the improvement. 

Overall the device is yet simple and low cost, but it is designed cleverly, and its implementation is clean. I could not find any flaw in the design and its working principle. One minor concern is the existence of hysteresis even after the refinement; which seems to be quite high (up to 8mN) considering the max adhesion force was measured only 25 mN. Maybe applying some lubricant to the pin may help.

The presentation of the experiment is a bit slack, which should be improved before the publication.

# Experiment 1)

The authors said they tested various substances, but only one (honey) was presented. I wonder what the authors expected when they designed the experiment with the multiple materials, which is missing in the result report.

# Experiment 2)

As the paper is discussing an assessment of a new sensing mechanism, a robust validation should follow. I liked that the authors used standard materials for the testing. However, the result report was insufficient. As the standard materials come with know properties, there must be followed by a comparison between the expected values and the measured values. For example, with a kinematic viscosity value A, the peak adhesive force B is expected, and the measured peak force was C, which is differed by (B-C) amount. The current result lack this validation. The result (Fig13) only says there is a correlation between the sensed value and the viscosity of substances, which is not sufficient to validate the accuracy and precision of the sensor. Also, please discuss why there was no difference between 5000cSt and 10000cSt conditions. Is it the limit of the sensor? Or is it the limit of the measuring method? Or is it the expected properties from the viscosity level? The second experiment has no additional support for the claim, "this noise is reduced" (line 286). Also, there are too few samples (only three) to conclude the reduction in noise level between the first and the second prototypes.

Author Response

Response to the comments of Reviewer 2

Comments and Suggestions for Authors

The authors designed a stickiness measuring apparatus consist of a load-cell array and pre-loading mechanism using magnets. The key idea is applying positive offset force to measure the negative pull from adhesion. The clear advantage of having a 2D array of sensors is the ability to detect force distribution.

I like the design, working principle, and the fabrication of the sensor. However, the validation experiments have a room for the improvement.

Overall the device is yet simple and low cost, but it is designed cleverly, and its implementation is clean. I could not find any flaw in the design and its working principle. One minor concern is the existence of hysteresis even after the refinement; which seems to be quite high (up to 8mN) considering the max adhesion force was measured only 25 mN. Maybe applying some lubricant to the pin may help.

The presentation of the experiment is a bit slack, which should be improved before the publication.

# Experiment 1)

The authors said they tested various substances, but only one (honey) was presented. I wonder what the authors expected when they designed the experiment with the multiple materials, which is missing in the result report.

Thank you for your comments. We added measurement results using toothpaste, shaving gel and shampoo.We added description about this in lines 241 to 243, 269 to 274.

# Experiment 2)

As the paper is discussing an assessment of a new sensing mechanism, a robust validation should follow. I liked that the authors used standard materials for the testing. However, the result report was insufficient. As the standard materials come with know properties, there must be followed by a comparison between the expected values and the measured values. For example, with a kinematic viscosity value A, the peak adhesive force B is expected, and the measured peak force was C, which is differed by (B-C) amount. The current result lack this validation. The result (Fig13) only says there is a correlation between the sensed value and the viscosity of substances, which is not sufficient to validate the accuracy and precision of the sensor. Also, please discuss why there was no difference between 5000cSt and 10000cSt conditions. Is it the limit of the sensor? Or is it the limit of the measuring method? Or is it the expected properties from the viscosity level? The second experiment has no additional support for the claim, "this noise is reduced" (line 286). Also, there are too few samples (only three) to conclude the reduction in noise level between the first and the second prototypes.

Thank you for your comments. The purpose of experiment 2 is to show that the system can grasp difference among different standard viscosity liquids, since honey, tooth paste, etc. that were used previously are daily substances that has little repeatability. Although we cannot say what physical property we grasped, we believe we can still say that the system has the ability to find difference among those standard liquids. The peak viscosity of 5000 cSt and 10000 cSt was the same not because of the limit of the sensor. We think it is the property of adhesive sample. Also, we think the higher kinematic viscosity, the stronger force generated on the pin when the adhesive sample enters the gap between the pins, has an effect on the two measurement results being the same.

Reviewer 3 Report

This work presents the use of a magnetic force to bias a pressure sensor array in order to detect tensile forces generated by contact with a sticky surface. There is some novelty here as most adhesion studies do focus on human perception, rather than the use of actual sensors, as generally in sensors adhesion is a factor to be considered, rather than a sensing modality. Adhesion or sktickness sesnors do exist however, such as AFMs, and related sensors, i.e. Liu, Huiwen, and Bharat Bhushan. "Adhesion and friction studies of microelectromechanical systems/nanoelectromechanical systems materials using a novel microtriboapparatus." Journal of Vacuum Science & Technology A: Vacuum, Surfaces, and Films 21.4 (2003): 1528-1538. It should also be noted that, as stated, the use of pressure sensors to measure stickiness has already been reported by the authors, therefore, it is only the use of magnetic forces to induce a bias in the sensors that is novel. 

Specific comments

The literature review is a little weak. There have been far more studies that is suggested here, i.e. Fisher, Tom H. "What we touch, touches us: Materials, affects, and affordances." Design Issues 20.4 (2004): 20-31. Chen, Xiaojuan, et al. "Exploring relationships between touch perception and surface physical properties." International Journal of Design 3.2 (2009): 67-76. Kimura, Fuminobu, Akio Yamamoto, and Toshiro Higuchi. "Development of a contact width sensor for tactile tele-presentation of softness." RO-MAN 2009-The 18th IEEE International Symposium on Robot and Human Interactive Communication. IEEE, 2009. ..., for instance. Therefore, a few more references are recommended. Reference error on line 155 needs correcting. The data in section 2.4 makes very little sense. There seems to be a lot of noise and variability. There is no sense of error or repeatability. Those in section 3.3 are only marginally better. It may be worth focusing on the improved design and spend more effort in assessing the quality of the data generated. There is no independent validation for the data in Fig. 13. It is also unclear whether this adhesion due to just viscous forces, such as squeeze flow, or due to surface tension or capillary forces, as there has been no attempt to analysis the data or give a theoretical background. 

Author Response

Response to the comments of Reviewer 3

This work presents the use of a magnetic force to bias a pressure sensor array in order to detect tensile forces generated by contact with a sticky surface. There is some novelty here as most adhesion studies do focus on human perception, rather than the use of actual sensors, as generally in sensors adhesion is a factor to be considered, rather than a sensing modality. Adhesion or sktickness sesnors do exist however, such as AFMs, and related sensors, i.e. Liu, Huiwen, and Bharat Bhushan. "Adhesion and friction studies of microelectromechanical systems/nanoelectromechanical systems materials using a novel microtriboapparatus." Journal of Vacuum Science & Technology A: Vacuum, Surfaces, and Films 21.4 (2003): 1528-1538. It should also be noted that, as stated, the use of pressure sensors to measure stickiness has already been reported by the authors, therefore, it is only the use of magnetic forces to induce a bias in the sensors that is novel.

Specific comments

The literature review is a little weak. There have been far more studies that is suggested here, i.e. Fisher, Tom H. "What we touch, touches us: Materials, affects, and affordances." Design Issues 20.4 (2004): 20-31. Chen, Xiaojuan, et al. "Exploring relationships between touch perception and surface physical properties." International Journal of Design 3.2 (2009): 67-76. Kimura, Fuminobu, Akio Yamamoto, and Toshiro Higuchi. "Development of a contact width sensor for tactile tele-presentation of softness." RO-MAN 2009-The 18th IEEE International Symposium on Robot and Human Interactive Communication. IEEE, 2009. ..., for instance. Therefore, a few more references are recommended. Reference error on line 155 needs correcting. The data in section 2.4 makes very little sense. There seems to be a lot of noise and variability. There is no sense of error or repeatability. Those in section 3.3 are only marginally better. It may be worth focusing on the improved design and spend more effort in assessing the quality of the data generated. There is no independent validation for the data in Fig. 13. It is also unclear whether this adhesion due to just viscous forces, such as squeeze flow, or due to surface tension or capillary forces, as there has been no attempt to analysis the data or give a theoretical background. 

Thank you for your comments. We added these references in lines 36, 44 to 46, 50 to 54. We fixed this reference error. Section 2.4 is certainly not very useful data, but we still would like to keep it to compare the result with those in Section 3.3.

 The purpose of measurement shown in Figure. 13 (16) is to show that the system can grasp difference among different standard viscosity liquids, since honey, tooth paste, etc. that were used previously are daily substances that has little repeatability. Although we cannot say what physical property we grasped, we believe we can still say that the system has the ability to find difference among those standard liquids.

Round 2

Reviewer 3 Report

Minor corrections have been made.

This manuscript is a resubmission of an earlier submission. The following is a list of the peer review reports and author responses from that submission.

Round 1

Reviewer 1 Report

(I received two versions of your article for review - one with yellow highlights and one without. The two editions had unfortunately slightly different content. My following comments are for the highlighted edition)

In the conclusion section you claim the sensor be highly accurate (line 281), although I am missing proof of that in the article. For example, you could validate your claim by comparing your sensor to an (industrial grade) calibrated load cell. When attached to your sensor, you could quantitatively evaluate the performance of pushing as well as pulling forces. As already previously complained, measurements using honey, baby-power etc. are interesting, but scientifically less sound.

The SI-unit for force is Newton. Please use it throughout the paper instead of "gf".

* Figure 2. the dimensions of the board are given to get a rough image about the size of the PCB. The printed precision with two digits after comma is in my opinion making reading cumbersome, thus I would suggest 10.2 times 36.8 mm, or even just 10 times 37 mm.

* Lines 94 to 96. The friction depends heavily on the chosen materials and their surface handling. You could discuss if you have considered alternative materials to acrylic.

* Line 99: what do you mean with "by 2"?

* Line 103: Please list your window size for the moving average.

* Lines 105 to 106: consider explicitly mentioning that laser cutter holes are typically slightly conical.

* Lines 109 to 111: Please validate if the sentence is only valid if all the magnets are at the same level/height.

* Line 113: what do you mean with "when"?

* Line 116: the mN value is listed with too high precision. Listing it as roughly 0.94mN would be better to read.

* the same line: the output raw value of 700 can be difficult to understand. You should not assume the reader to know, or have to look it up, that MCP3208 is a 12-bit ADC.

* still the same line: Why the preload used was only 700? Wouldn't a higher preload value (in the 12-bit range of 0 to 4095) be better for adhesion measurements?

* Figure 4: The distance value between the acrylic plates would also be interesting to read.

* Figure 5 caption needs rephrasing, as not only a single circuit board is shown.

* Figure 6 caption: "based on magnet" should be rephrased. And if you desire to reference figure 4 in the caption, it might be better to combine figures 4 and 6.

* Figure 7: Consider exchanging horizontal and vertical axes. Typically the driving value is plotted in the horizontal axis. In the calibration case that would be the weight applied.

* Line 117: The reference should be to figure 8 instead of to figure 7.

* Line 118: inconsistency between figure 8 caption and the line here. Fig. 8 says "immediately at the surface of the magnet", the sentence here says "above the substrate". Which is correct?

* Line 148: please explain more clearly what do you mean by "100 times".

* Lines 148 and 151: please minimally list the manufacturer and type of baby powder used.

* Lines 168, 171 and 248: "Newton" with capital N. Also "The horizontal plane represents..."

* Please explain the meaning of the 4-digit values in the diagram titles in figures 9 and 10, or consider omitting them similar to figure 15.

* Why figure 11 lists only one acrylic plate, whereas figure 4 lists two?

* Lines 176 & 194: "Measurement System..." & "Measurement Device..."

* Line 203 lists 3 magnets, line 204 talks about two base magnets. Please check if this is consistent/correct.

* Line 221: please further explain the sentence starting with "We placed...". I cannot unfortunately follow what you did from the description.

* Have you considered rectangular pin magnets to have less material creep/flow between the magnets?

* Line 289: likely "depressed" needs to be changed to "pressed".

Reviewer 2 Report

This paper developed and evaluated a device for stickiness sensing upon touching an adhesive substance.  An offset pressure was applied to load cell pressure sensor.  This offset pressure was generated by using a magnetic force to control magnetic pin arrays and pin magnets.  Several demonstrations have been implemented to prove this concept.  However, several major issues, listed below, that make this manuscript below the standard level for publications in Micromachines.

The innovation of this work is limited. The load cell, the key component of this sensing system, was bought from market.  Similar designs of magnetic pins have also been published by researchers in other applications.    A lot of details are missing. For example, what is the part number of the circuit board shown in Figure 3?  What are the components on it?  Are they the same as the components in Figure 5?  What is the reason that a different board is used for the final equipment in Figure 5?  Is it customized?  How is this board shown in Figure 5 connected to the magnet pins?  The authors did not provide any 3D drawing of the magnetic pins, which is the most important innovation of this work, and the assembling of the system is not clearly presented. The way that the authors present results in Figure 8, 9, and 10 are confusing. It is hard for readers to understand the meanings behind these figures, and the authors failed to provide sufficient explanations. In Section 3, the authors made modifications to the system to solve the problems of the prototype in Section 2. However, the demonstration of the improved system is different from the experiment done in the original prototype.  So, it is hard to make a solid conclusion for improvement by using two different validating method. The quality of presenting, graphs, and analysis is below the average standard.